# An Improved Instance Segmentation Method for Fast Assessment of Damaged Buildings Based on Post-Earthquake UAV Images

**DOI:** 10.3390/s24134371

**Published:** 2024-07-05

**Authors:** Ran Zou, Jun Liu, Haiyan Pan, Delong Tang, Ruyan Zhou

**Affiliations:** 1School of Information Science, Shanghai Ocean University, Shanghai 201306, China; 2152730@st.shou.edu.cn (R.Z.); hy-pan@shou.edu.cn (H.P.); ryzhou@shou.edu.cn (R.Z.); 2National Earthquake Response Support Service, Beijing 100049, China; 3Guizhou Provincial Seismological Bureau, Guiyang 550001, China; dltang@gzsdzj.gov.cn

**Keywords:** earthquake rescue and assessment, build damage classification, UAV images, instance segmentation, YOLOv5-Seg

## Abstract

Quickly and accurately assessing the damage level of buildings is a challenging task for post-disaster emergency response. Most of the existing research mainly adopts semantic segmentation and object detection methods, which have yielded good results. However, for high-resolution Unmanned Aerial Vehicle (UAV) imagery, these methods may result in the problem of various damage categories within a building and fail to accurately extract building edges, thus hindering post-disaster rescue and fine-grained assessment. To address this issue, we proposed an improved instance segmentation model that enhances classification accuracy by incorporating a Mixed Local Channel Attention (MLCA) mechanism in the backbone and improving small object segmentation accuracy by refining the Neck part. The method was tested on the Yangbi earthquake UVA images. The experimental results indicated that the modified model outperformed the original model by 1.07% and 1.11% in the two mean Average Precision (mAP) evaluation metrics,$\;{mAP}_{bbox}^{50}$ and ${mAP}_{seg}^{50}$, respectively. Importantly, the classification accuracy of the intact category was improved by 2.73% and 2.73%, respectively, while the collapse category saw an improvement of 2.58% and 2.14%. In addition, the proposed method was also compared with state-of-the-art instance segmentation models, e.g., Mask-R-CNN and YOLO V9-Seg. The results demonstrated that the proposed model exhibits advantages in both accuracy and efficiency. Specifically, the efficiency of the proposed model is three times faster than other models with similar accuracy. The proposed method can provide a valuable solution for fine-grained building damage evaluation.

## 1. Introduction

Earthquakes, one of nature’s most catastrophic occurrences, can seriously harm a building’s structural integrity [1]. Building damage must be classified promptly and accurately in order for government rescue and emergency response efforts to be successful. Building damage evaluation benefits greatly from the extensive coverage, high resolution, and excellent timeliness of remote sensing images [2,3].

Building extraction and classification from high-resolution, pre-disaster remote sensing images has long been a major area of interest for both industry professionals and academics. Classical techniques for extracting buildings from remote sensing images can be divided into two primary types based on feature extraction: data-driven techniques and model-driven techniques [4]. Data-driven approaches identify building targets based on specific combination rules by primarily taking into account low-level information seen in remote sensing photos, such as lines, corners, texture regions, shadows, and height disparities. Model-driven approaches use some high-level global features to direct processes, including item extraction, image segmentation, spatial connection modeling, and contour curve development towards building targets. These approaches begin with the semantic model and a pre-existing understanding of the aim of the entire building.

As artificial intelligence and computer vision have advanced, deep learning has gained popularity as a method for acquiring high-level features that are both discriminative and representative [5]. Deep learning is more flexible and capable than conventional techniques since it can directly learn high-level characteristics from raw data [6]. Convolutional Neural Networks (CNNs) have been extensively utilized in the last few years to recognize damaged buildings [7].

Some scholars have conducted earthquake-damaged house classification research using deep learning methods. XView2 is a large-scale remote sensing image dataset used for building detection and classification in natural disasters. This dataset provides high-resolution aerial images before and after disasters, classifying various damage types into four categories, namely no damage, minor damage, major damage, and destroyed. For example, Weber et al. (2020) used Unet on the Xiew2 dataset to extract the pre-disaster and post-disaster image features of buildings. They located buildings based on pre-disaster images and classified the damage categories of buildings based on post-disaster images. The overall F1 score was a weighted combination of 30% localization F1 and 70% damage classification F1, achieving an overall F1 score of 74.1% [8]. There are also scholars who divide the disaster level into two categories, namely collapsed and not collapsed. Wang et al. (2021) proposed an improved network, OB-UNet, and built two benchmark datasets (YSH and HTI) for identifying damaged buildings based on post-earthquake images from China and Haiti in 2010, obtaining a mIoU of 66.77% on the YSH dataset and 70.95% on the HTI dataset [9]. Cui et al. (2022) proposed an improved Swin Transformer method, achieving a mIoU of 88.53% [10]. The aforementioned method primarily utilizes semantic segmentation. Additionally, some scholars divide post-disaster images into multiple smaller sub-images and classify damage within each sub-image. In this approach, the entire covered area within each sub-image is identified as a single damage level, and the results are then assembled to achieve an overall assessment of the area’s damage. For example, Cooner et al. (2016) used CNNs to classify high-resolution earthquake remote sensing images, dividing the disaster level into two categories, namely damaged and undamaged, quickly detecting damaged buildings, with an accuracy rate of 55% for the 2010 Haiti 7.0 magnitude earthquake [11]. Ma et al. (2020) combined remote sensing images with block vector data, divided the disaster level into three categories, serious damage, moderate damage, and slight damage, and improved the Inception V3 architecture; the test precision for post-earthquake aerial images of the Yushu earthquake reached 90.07% [12]. Ji et al. (2020) used a pre-trained VGG model to identify collapsed buildings in pre- and post-earthquake remote sensing images of the 2010 Haiti earthquake, dividing the disaster level into two categories, namely collapsed and not collapsed. The results showed that the adjusted VGGNet model outperformed the original VGGNet model trained from scratch, improving the overall accuracy from 83.38% to 85.19% [13]. In addition, some scholars have used target detection methods to solve this problem. Gadhave et al. (2023) used YOLOv8 on the xView2 dataset for building detection and classification (four categories) tasks, achieving the best ${mAP}^{50}$ of 58.3% [14]. Jing al. (2022) used an improved YOLOv5 on the Yunnan Yangbi dataset for collapsed building detection tasks, reaching ${mAP}^{50}$ of 90.94% [15]. Wang et al. (2023) proposed an improved BD-YOLOv5 for collapsed building detection tasks, improving the F1 score from 94.08% to 95.34% [16].

In past research, the method of dividing images into smaller patches and classifying each sub-image has been utilized, and each sub-image region is identified as a single damage level. Its advantage lies in achieving higher accuracy. However, in high-resolution Unmanned Aerial Vehicle (UAV) imagery, this method may split a single building across different sub-images, resulting in multiple damage states for a single target. Additionally, for irregularly shaped buildings, this approach may struggle to accurately determine building edges and can only provide a rough estimation of the building’s damage status. Figure 1b illustrates this effect. The object detection method cannot accurately extract building outlines as it can only predict the location of damaged buildings, treating the entire predicted bounding box as a single instance of the damaged building area (Figure 1c). When the shape of the building is irregular, this method tends to overestimate the predicted damaged area, which is not conducive to performing a fine-grained assessment after an earthquake. Semantic segmentation, which has had positive results in earlier research, is generally the foundation of current deep learning techniques for extracting buildings with varying degrees of damage from high-resolution remote sensing images. However, in complex geographical contexts and low-altitude UAV data, where targets are much larger compared to satellite imagery, semantic segmentation may lead to edge connections between adjacent buildings. Confusion regarding the building boundaries may be the cause of this, which is detrimental to edge extraction and contour fitting later on [17]. Additionally, semantic segmentation, which classifies each pixel, may result in multiple classification results for different parts of the same target building. In principle, each building instance should only have one state. However, many weaker disasters result in partial damage to buildings, with damaged areas often much smaller than undamaged areas. Undamaged samples still dominate the learning process, causing pixel-level models to segment damaged building instances into several different area categories [8,18], as shown in Figure 1d.

Instance segmentation encompasses object detection and semantic segmentation, allowing for the precise identification and differentiation of various building instances within an image. It also accurately delineates the boundaries of each building instance. Mask R-CNN [19], a trailblazing study in instance segmentation that involves simultaneously predicting segmentation masks and bounding boxes, has made notable progress. Currently, according to the mask segmentation principle, instance segmentation is mainly divided into two categories: single-stage and two-stage. Representative two-stage methods include Mask R-CNN, Cascade Mask R-CNN [20], and HTC (Hybrid Task Cascade for Instance Segmentation) [21]. Single-stage methods include YOLACT [22], CondInst [23], SOLO [24], SOLOv2 [25], YOLOv5-Seg, and YOLOv8-Seg. When it comes to real-time performance and detection speed, single-stage detection methods outperform two-stage methods. For example, Yildirim et al. (2023) applied Mask-R-CNN to remote sensing data collected after the Kahramanmaraş earthquake in Türkiye. They categorized the disaster level into two classes, namely intact and collapsed. Experimental results showed that the Mask R-CNN model with a ResNet-50 backbone produced the most accurate results, successfully distinguishing intact and collapsed buildings, with average precisions (AP) of approximately 81% and 69%, respectively [26]. Zhan et al. (2022) proposed an improved Mask-R-CNN for a dataset of earthquakes in the Kyushu region of Japan in April 2016, categorizing the disaster level into four categories (no damage, slight damage, severe damage, and collapsed). They achieved ${mAP}_{bbox}^{50-95}$ improvements from 33.2% to 36.5% and ${mAP}_{seg}^{50-95}$ improvements from 33.3% to 37.3% [27].

In the research based on post-earthquake UAV images, instance segmentation methodology stands out for its ability to meet the diverse data requirements of post-earthquake rescue efforts compared to other research methods. Not only does it enable the extraction and classification of buildings at different damage levels from images, but it also allows for the counting of individual buildings and the differentiation between different instances of buildings. This approach is better suited for the fine-grained needs of earthquake damage assessment and rescue operations. However, as far as the authors are aware, there has not been much research on instance segmentation in building damage categorization. [17].

Based on the research of domestic and international scholars, this paper proposes an improved YOLOv5s-Seg algorithm for extracting buildings at different damage levels from post-earthquake UAV images. This paper’s primary innovations are:This paper is different from most of the existing studies that use semantic segmentation or target detection methods for building damage classification. This paper adopts the instance segmentation method to classify the damage level of buildings after the earthquake when dealing with UAV imagery, which can generate more precise acquisition of building locations and damaged areas.To address the issue of classification errors between adjacent categories of damaged buildings due to similar texture features, such as between the Intact and Slight categories and the Severe and Collapse categories, this paper has incorporated the MLCA (Mixed Local Channel Attention) mechanism into the backbone. This combines local and global features, as well as channel and spatial feature information, enhancing image feature extraction and thus improving the accuracy between the Intact and Collapse categories.In order to enhance the segmentation precision of small buildings, this paper replaced the Neck part with the Neck part from ASF-YOLO (A Novel YOLO Model with Attentional Scale Sequence Fusion). This Neck part can strengthen the feature information of small targets, thereby improving the model’s detection and segmentation precision.

## 2. Materials and Methods

### 2.1. Data Acquisition

In order to confirm the efficacy of the suggested approach, this paper utilized a dataset of UAV images taken after the Yangbi earthquake in Yunnan, China. On 21 May 2021, a Ms6.4 earthquake struck Yangbi County, Yunnan Province, China. Figure 2 shows the location of the epicenter, which was 8 km deep and at (25.67° N, 99.87° E). At its strongest, this earthquake had an area of around 170 square kilometers and was felt most strongly in the cities of Cangshanxi, Yangjiang, and Taiping in Yangbi County. Its maximum intensity was VIII. Due to economic constraints and weak earthquake preparedness, houses in these rural areas had lower seismic resilience [28,29,30]. Consequently, some houses in intensity zones VII–VIII suffered severe damage or collapsed. According to local government statistics, 232 rooms collapsed and 13,930 rooms experienced moderate to slight damage [31]. The main types of building structures in this area include civil, brick–wood, and brick–concrete structures [32].

Shanghai Ocean University and the Chinese Academy of Sciences’ Aerospace Information Research Institute (AIR/CAS) acquired the UAV data. DJI Phantom 4 collected post-earthquake UAV images from 23–25 May 2021, at flight altitudes of 50–200 m and spatial resolutions of 0.01–0.06 m. Due to the continuous nature of the original image data and the high overlap between adjacent images, for preparing training samples, we selected 200 distinct images from a pool of 2235 orthoimage samples. Each image sample includes multiple damaged building areas. The preprocessing steps included data augmentation, annotation of damaged building areas, and classification. In this study, to preserve as many image details as possible, we segmented the original images into 1024 * 1024-sized patches. The model input image size was 1024 * 1024 pixels, ensuring that each sub-image contained at least one damaged building area.

For aerial UAV remote sensing orthoimages, building damage levels can be classified based on the roofs and surroundings of buildings [32]. Currently, the degree of earthquake damage at the seismic site is typically characterized using a seismic damage index. Based on the current “Chinese Seismic Intensity Scale” (GB/T 17742-2020 [33]), building damage levels are divided into five categories (basically intact, slight damage, moderate damage, severe damage, and destruction), with corresponding definitions of seismic damage indices. Due to limitations in image resolution and observation distance, it was challenging for the authors of this study to obtain complete damage differentiation for slight and moderate damage. As a result, this paper offers four categories of building damage for UAV images, including basically intact buildings, slightly damaged buildings, severely damaged buildings, and completely collapsed buildings. They are described as follows in Table 1.

In this paper, the open-source program labelme was used to annotate the damaged parts of buildings. As illustrated in the image, green boxes were utilized to annotate the damaged building areas labeled with labelme. The annotation results were saved as a json file. Below are the annotations and visualization results (Figure 3).

### 2.2. Image Preprocessing

Based on the created dataset, we employed data augmentation to produce additional building images with different degrees of damage. However, the augmentation techniques for natural and remote sensing (RS) images differ in a few ways. For instance, structures in this study can rotate at any angle, whereas most things in natural images usually only rotate at tiny angles. The primary augmentation processes are picture rotation and both vertical and horizontal flipping in order to account for the various viewing angles. The brightness of the images varied by 20% as a result of changes in the observation periods and environmental factors. A few examples of common augmentation processes are shown in Figure 4.

After applying the above-indicated data augmentation techniques, a total of 5394 images were obtained. The specific dataset divisions and examples of each class are shown in Table 2.

### 2.3. Model Training and Performance Evaluation

An NVIDIA GeForce RTX 3090 graphics card (24576MiB, provided by ASUS, Shanghai, China) and a 12th Gen Intel(R) Core (TM) i9-12900K CPU @ 3.20 GHz (provided by ASUS, Shanghai, China) powered the entire experimental setup, which ran on Windows 11. PyTorch 2.0.1, the deep learning framework, and CUDA 11.7, the Compute Unified Device Architecture, were the versions utilized. The details of the experimental environment are summarized in Table 3.

Consistent experimental parameters, including the image input size set to 1024 * 1024 pixels, training epochs set to 150, SGD optimization, and batch size fixed at 10, were maintained to guarantee impartial and equitable comparisons across all trials. Table 4 displays the experimental parameters. Additionally, during testing and evaluation, we set the Non-Maximum Suppression (NMS) value for bounding boxes to 0.7. It is worth noting that due to significant differences in the principles, methods, and structures of different models, other hyperparameters are not entirely transferable. To ensure fair and just comparative experiments, a comparative experimental model was trained using multiple sets of hyperparameter combinations, and the best results were selected for the final comparison.

This paper evaluates the model performance of the improved YOLOv5-Seg using precision, recall, and the standard COCO instance segmentation metric of mean Average Precision (mAP) [34]. The localization of buildings is evaluated using ${mAP}_{bbox}^{50}$, while the segmentation results are evaluated using ${mAP}_{seg}^{50}$.

In instance segmentation, Intersection over Union (IoU) represents the degree of overlap between the predicted bounding box or mask by the model and the ground truth label. For bounding boxes (bbox), the calculation involves two rectangular areas; in the case of masks, it involves each pixel within the regions. The IoU value is obtained by dividing the area of the intersection by the area of the union. The formula is expressed as follows:(1)$$IoU=\frac{Intersection\;area}{Union\;area}$$

The equations used to calculate precision, recall, and mAP scores are as follows. Higher values indicate better segmentation results.
(2)$$P=\frac{TP}{\left(TP+FP\right)}\times 100\%,$$
(3)$$R=\frac{TP}{\left(TP+FN\right)}\times 100\%,$$
(4)$$AP=\int_{0}^{1}P\left(R\right)dR,$$
(5)$$mAP=\sum_{i=1}^{c}\frac{AP\left(i\right)}{C}$$

The range of IoU values is from 0 to 1, where 0 represents no overlap and 1 represents complete overlap. As forecasts with an IoU > 0.5, TP stands for the number of true positive predictions. When a prediction has an IoU ≤ 0.5, it is considered a false positive, and its number is represented by FP. When real positive samples are predicted to be negative, the number of false negative predictions, or FN, is represented. AP denotes the average precision of segmentation. The segmentation performance of the model increases with higher AP scores. In the context of these calculations, C represents the number of classes, P represents precision, and R represents recall.

One important performance indicator in the model is frames per second (FPS), which is used to express how many images the model can handle in a second. The reciprocal of the processing time (measured in seconds) for one frame is used to determine frames per second (FPS). The calculation method is as follows:(6)$$FPS=\frac{1}{T}\;$$

### 2.4. Improved YOLOv5-Seg Model

YOLOv5-Seg is based on YOLOv5. YOLOv5, a deep learning-based object detection algorithm, uses effective reasoning algorithms and a lightweight network structure to significantly increase detection speed without sacrificing accuracy. By improving and expanding YOLOv5, it can be applied to image segmentation tasks.

By analyzing the network structure of YOLOv5-Seg, the method proposed in this paper has made improvements in its Backbone and Neck parts, including adding an attention mechanism, MLCA, to the backbone to aggregate similar features for improving the model’s precision on house damage level I and IV categories. The Neck part is replaced by the Neck part in ASF-YOLO, a network used for small target detection, combining global or high-level semantic information from feature maps of different sizes, thereby improving the detection and segmentation accuracy of small targets. The improved network structure is shown in Figure 5. The detailed module structure is shown in Figure 6, and the following sections will sequentially introduce the specific principles and implementations of each module.

#### 2.4.1. Mixed Local Channel Attention (MLCA)

The building damage in the Collapse and Severe categories, as well as the Intact and Slight categories, might be difficult to distinguish from one another because of their comparable features. The main differences in the destruction of buildings are primarily observed on the rooftops, with variations in the extent of damage. Therefore, it is necessary to mix similar qualities in order to improve the separation of different levels of building damage [35]. Furthermore, the structures’ surrounding texture and geographical information might offer crucial additional data for evaluation. The accuracy of the model’s classification of the degree of building damage can be improved by investigating the interaction between global and contextual variables inside images.

This study proposes integrating an MLCA attention mechanism into the backbone as a solution to these problems. This attention mechanism can effectively adjust the model’s feature weight ratios for building outlines, rooftops, and wall features, enabling local perception to focus on different positions of input data to varying degrees. It also uncovers correlations and dependencies between different channels in the data, adjusting the attention weights between channels. By doing so, the model can make more effective use of information between channels, thereby enhancing the overall performance and effectiveness of the model.

MLCA (Mixed Local Channel Attention) [36] is a mixed local channel lightweight attention mechanism that combines local and global features, as well as channel and spatial feature information. With just a slight increase in the number of parameters, detection accuracy can be greatly improved. Figure 7 shows the schematic diagram, and the following are the implementation steps:

Assuming the input data is X with shape (N,C,H W), first perform local average pooling operation on the input data X. Let $U_{0}$ represent the output of the pooling result, and $K_{s}$ represent the size of the feature map after pooling. After local average pooling, obtain the feature vector $U_{0}$ with shape (N, C, $K_{s}$, $K_{s}$).
(7)$$U_{0}=P_{LAP}\left(X\right)$$

Building upon this foundation, two branches are utilized to extract different feature information. The first branch $S_{1}$ encompasses global information, while the second branch $S_{2}$ encapsulates local spatial information.

The processing of $S_{1}$ is as follows: the input undergoes another global average pooling, resulting in a shape of (N,C,1,1). Then, it is reshaped to (N,1,C) and processed through Conv1d. Subsequently, reshaping is performed again, resulting in a shape of (N,C,1,1). The results are passed through a sigmoid activation function and then undergo reverse average pooling to match the shape of $S_{2}$, resulting in the output $U_{1}$ with a shape of (${N,C,K}_{s},K_{s}$).
(8)$$V_{1}=F^{conv1d}\left(F^{reshape}\left(P_{GAP}\left(U_{0}\right)\right)\right),$$
(9)$$U_{1}=P_{UNAP}\left(\sigma \left(F^{reshape}\left(V_{1}\right)\right)\right)$$

The processing of $S_{2}$ is as follows: the input is reshaped to ($N,1,1,C*K_{s}*K_{s}$), then processed through Conv1d. Subsequently, reshaping is performed again, resulting in a shape of (${N,C,K}_{s},K_{s}$). Finally, a sigmoid activation function is applied to obtain the output $U_{2}$.
(10)$$V_{2}=F^{conv1d}\left(F^{reshape}\left(U_{0}\right)\right)\;,$$
(11)$$U_{2}=\sigma \left(F^{reshape}\left(V_{2}\right)\right)$$

The outputs of the two branches are then weighted and summed. Subsequently, reverse average pooling is performed to restore the feature map size to its original dimensions (N,C,H W). Lastly, the final output $U_{3}$ is obtained by multiplying the result element-wise with the original input, where ∂ is the weighting coefficient, with default set to 0.5.
(12)$$U_{3}={X\cdot P}_{UNAP}\left({\partial \cdot U}_{1}+{\left(1-\partial \right)\cdot U}_{2}\right)$$

In the above formulas, $P_{LAP}$ represents local average pooling, $P_{GAP}$ represents global average pooling, and $P_{UNAP}$ represents anti-pooling, which can be achieved through adaptive average pooling, $F^{reshape}$ represents reshaping the feature map, σ is the sigmoid activation function, and $F^{conv1d}$ denotes one-dimensional convolution. The channel diameter (c) is directly proportional to the kernel size (k), indicating that when capturing local cross-channel interaction information, only the relationships between each channel and its k adjacent channels are considered. The selection of k is determined by the following formula [37]:(13)$$k=\varphi (c)=|\frac{\log_{2}\left(c\right)}{\gamma }+\frac{b}{\gamma }{|}_{odd}$$

c is the number of channels, k is the size of the convolutional kernel, γ and b are hyperparameters, with γ defaulting to 2 and b defaulting to 1. If k is even, add 1.

#### 2.4.2. Attentional Scale Sequence Fusion (ASF)

For aerial UAV images, the number of small objects in the image increases with the flight altitude. Consequently, as the flight altitude rises, the model’s ability to extract features related to damaged buildings decreases, resulting in higher rates of missed detections and reduced segmentation accuracy. To address this issue, this paper introduced the Neck component from ASF-YOLO [38] into the Neck part of YOLOv5-Seg, thereby enhancing the model’s ability to extract features from small objects and consequently improving the overall detection and segmentation accuracy of the model.

1.Overall Architecture

The Neck part in Figure 5 presents an overview of the ASF framework, which combines spatial and multi-scale features, e.g., segmentation. The Triple feature encoding (TFE) module, which captures minute details of smaller buildings, and the Scale sequence feature fusion (SSFF) module, which combines global or high-level feature information from various scale images, are the two main component networks that make up this architecture and provide complementary feature information for small building segmentation. More accurate segmentation outcomes can arise from the efficient fusion of the image’s local and global feature information. For the YOLOv5 backbone network, the high-level feature information from P3, P4, and P5 is fused. Initially, the SSFF module efficiently merges the feature information from P3, P4, and P5. These three layers capture information about buildings of various sizes and shapes at different spatial scales. P3, P4, and P5 feature maps are processed to the same scale, up-sampled, and concatenated in the SSFF module before being used as input data for 3D convolution, which efficiently combines features from various scales. The purpose of the TFE module is to improve the network’s capacity to identify small buildings in crowded metropolitan environments. It captures detailed information about small buildings by combining three different-sized feature maps in the spatial dimension. Subsequently, the TFE module’s data is included into every feature branch via the Path Aggregation Network (PANet) structure, merging with the multi-scale data from the SSFF module to create the P3 branch. In addition, a channel and position attention mechanism (CPAM) is added to the P3 branch in order to take advantage of detailed features and high-level multi-scale characteristics. By refining the spatial placement of tiny buildings and capturing information channels, the position attention and channel processes in CPAM enhance the accuracy of small building detection and segmentation.

2.Scale sequence feature fusion module (SSFF)

The SSFF module is designed based on the P3 level of the high-resolution feature map since it provides the majority of the data required for small object recognition and segmentation. The specific structure is illustrated in Figure 8. The implementation details of the SSFF module are as follows: Utilize 1 × 1 2D convolution to adjust the channel numbers of layers P4 and P5 to match those of P3. Employ the nearest neighbor interpolation method to resize their feature map sizes to match that of the P3 layer. Each feature layer’s dimension is increased by using the unsqueeze method, which transforms it from a 3D tensor [height, width, channel] to a 4D tensor [depth, height, width, and channel]. Subsequently, 3D feature maps are created by concatenating the 4D feature maps along the depth dimension, ready for further convolution. Lastly, the SiLU activation function, 3D convolution, and 3D batch normalization are used to finish the scale sequence feature extraction process.

3.Triple feature encoding module (TFE)

Enlarging images and referring to and comparing shape or appearance variations at different scales can be a useful method for identifying densely overlapping small items. However, traditional Feature Pyramid Network (FPN) fusion mechanisms typically only upsample small-sized feature maps and then split or add them to the features of the preceding layer, thereby neglecting the rich detailed information in larger-sized feature layers because different feature layers in the backbone network have varying sizes. Detailed feature information can be improved by using the TFE module, which divides large, medium, and small features, adds large-sized feature maps, and carries out feature magnification.

The TFE module’s structure is shown in Figure 9. The number of feature channels is first changed to correspond to the medium-sized feature channels before feature encoding. After processing the large-sized feature map, the convolutional module sets the channel number to 1C. Next, downsampling is achieved using a hybrid structure that combines average and max pooling to preserve high-resolution features while guaranteeing effectiveness and diversity. The convolutional module is also used to modify the channel number for small-sized feature maps. This is followed by upsampling using the closest neighbor interpolation technique. This keeps feature information for small objects from being lost and helps maintain the richness of local features in low-resolution images. Finally, the feature maps of large, medium, and small sizes, which have been resized to the same dimensions, undergo another round of convolution separately. Subsequently, the results of these convolutions are concatenated together.
(14)$$F_{TFE}=Concat\left(F_{l},F_{m},F_{s}\right)$$
where $F_{TFE}$ represents the feature map outputted by the TFE module, $F_{l},F_{m}$, and $F_{s}$, respectively, denote the feature maps of large, medium, and small sizes. $F_{TFE}$ is generated by concatenating $F_{l},F_{m}$ and $F_{s}$.

4.Channel and Position Attention Mechanism (CPAM)

CPAM is used to integrate multi-scale and comprehensive feature information in order to derive representative feature information from many channels. In Figure 10, the CPAM architecture is shown. It consists of a channel attention network (${Net}_{1}$) that takes TFE input and a position attention network (${Net}_{2}$) that takes both channel attention network and SSFF output as input.

The detailed procedure for ${Net}_{1}$ implementation is as follows: for input data $S_{1}$, carry out global average pooling; after that, rework the output and send it through Conv1d. Finally, reshape it back to the original input shape to obtain output $U_{1}$. This process is used to effectively capture the relationships between different channels.

Next, the input data $S_{2}$ is added to $U_{1}$ to obtain E, forming the input data for ${Net}_{2}$.

The position attention mechanism divides the input feature map along the width and height dimensions first, then processes each component independently, in contrast to the channel attention method. Feature encoding is performed on the axes ($P_{w}$ and $P_{h}$), and finally, the outputs are merged.

More precisely, the information from the input feature map is aggregated along two different directions: horizontally ($P_{w}$) and vertically ($P_{h}$). This approach effectively preserves spatially structured relevant features of the feature map. The following formulas can be used to calculate it:(15)$$p_{W}=\frac{1}{H}\sum_{0\leq j\leq H}E\left(w,j\right),$$
(16)$$p_{h}=\frac{1}{W}\sum_{0\leq i\leq W}E\left(i,h\right)$$

The input feature map’s width and height are represented by the letters W and H, respectively. The values corresponding to the position (i,j) of the input feature map E are represented by the variables E(w,j) and E(i,h), respectively. The characteristics in the horizontal and vertical directions are concatenated first, and then a Conv2d operation is applied to generate the coordinates for position attention:(17)$$P\left(a_{w},a_{h}\right)=Conv2d\left[Concat\left(p_{w},p_{h}\right)\right]$$

The obtained attention features are further segmented to obtain feature maps associated with positions, as shown below:(18)$$S_{w}=Split\left(a_{w}\right),$$
(19)$$S_{h}=Split\left(a_{h}\right)$$
where $S_{w}$ and $S_{h}$, respectively, represent the feature maps on the width and height dimensions. Their shapes and dimensions are expanded to match E, then multiplied together to obtain the final output.

The definition of CPAM’s ultimate output is:(20)$$F_{CPAM}=E\times S_{\omega }\times S_{h}$$

#### 2.4.3. Loss Function

In earthquake events, due to differences in building structures across various regions and the varying types of earthquake-induced damage to buildings, there is an imbalance in the categories within the training sample data. Therefore, the focal loss function was adopted because it can effectively address the issue of category imbalance regarding different levels of building damage [39,40].

Although the traditional cross-entropy loss function is widely used, its performance often significantly declines when dealing with inter-class imbalance problems. Focal Loss is a loss function specifically designed to address category imbalance issues. The core idea is to introduce a modulation factor based on the standard cross-entropy loss. This factor can dynamically reduce the weight of samples that the model can easily classify, thereby enabling the model to focus more on those samples that are difficult to classify and improve the learning efficiency for hard samples.

The specific implementation formula for Focal Loss is as follows:(21)$$FL\left(P_{t}\right)=-\alpha_{t}{\left(1-P_{t}\right)}^{\gamma }\log\left(P_{t}\right)$$

$P_{t}$ is the probability of the model predicting a positive class sample, $\alpha_{t}$ is a weighting factor for different classes used to balance class imbalance, and γ is a modulation factor used to reduce the loss contribution of easily classified samples, commonly set to 2.

## 3. Results

YOLOv5-Seg offers several training models of varying sizes, n, s, m, l, and x, to satisfy the varied requirements of numerous application situations (full details as shown in Table 5). In particular, the smallest and fastest model is YOLOv5n-Seg. YOLOv5x-Seg, on the other hand, has the most parameters out of all the models and provides the best accuracy but at a slower execution pace.

Taking into account the demands of real-time tasks and device deployment for model parameters, this study especially focuses on YOLOv5s-Seg.

The comparative experimental results are shown in Table 6 and Figure 11. From the results data, it is evident that the improved YOLOv5s-Seg model exhibits certain advantages in terms of ${mAP}^{50}$ evaluation metrics, model parameters, and inference speed compared to other algorithms. In terms of the ${mAP}^{50}$ metric, both for bounding boxes (${mAP}_{bbox}^{50}$) and segmentation (${mAP}_{seg}^{50}$), the improved YOLOv5s-Seg model performs better than alternative methods. Most notably, the ${mAP}_{bbox}^{50}$ of YOLOv5s-Seg reaches 68.84%, which is 0.56% higher than the best comparative algorithm YOLOv9-Seg, and ${mAP}_{seg}^{50}$ is 0.15% higher. These results show that the modified approach considerably raises the model’s accuracy in identifying and classifying post-earthquake building damage. Furthermore, the improved YOLOv5-Seg model shows a notable edge over YOLOv7-Seg [41] and the most recent YOLOv9-Seg [42] when comparing the quantity of model parameters.

As can be seen in Figure 11, the model’s parameters are noticeably smaller than those of other models with a comparable level of precision. The parameter count of YOLOv5s-Seg is 7.66 M, which is less than half of YOLOv7-Seg’s (37.86 M) and YOLOv9-Seg’s (27.4M), with similar ${mAP}^{50}$. This indicates that the improved model maintains excellent accuracy while drastically reducing the computational burden and model size, thereby improving the model’s efficiency. Furthermore, with a comparatively minor increase in model parameters, the improved YOLOv5s-Seg model exhibits a notable improvement in ${mAP}^{50}$ when compared to the earlier YOLOv7-Seg and YOLOv9-Seg models, indicating the efficacy of the improvement technique. This benefit is ascribed to the suggested enhancement techniques, which include the enhanced Neck module and the MLCA attention mechanism. These improvement strategies have, to some extent, enhanced the model’s performance while controlling the increase in model complexity, increasing the model’s viability and usefulness in real-world scenarios.

When compared to previous models, the improved YOLOv5s-Seg model attains an optimal compromise between model accuracy and inference speed. As demonstrated in Table 6, the improved model outperforms other models of the same accuracy in terms of speed. It is worth noting that the improved YOLOv5s-Seg maintains a good FPS (Frames Per Second) value while retaining accuracy, reaching 142 in all experiments, which is approximately three times faster than models with similar accuracy, such as YOLOv9-Seg and YOLOv7-Seg. This means that the model has a significant advantage in real-time tasks, such as real-time detection of UAV videos during post-earthquake rescue missions.

Although the improved model in this paper is slightly slower compared to the original YOLOv5s-Seg model, its accuracy has been significantly improved. Researchers commonly think that in scenarios such as post-earthquake rescue and evaluation, a good model tries to achieve a compromise between fast detection speed and high accuracy. The goal of the improved YOLOv5s-Seg model presented in this work is to improve outcomes in both aspects.

## 4. Discussion

In order to methodically assess each augmentation strategy’s contribution to the YOLOv5-Seg model’s overall performance, ablation research was carried out in this section. The following summarizes the YOLOv5s-Seg models with different improvement strategies: MLCA and ASF. Table 7 lists the improvement tactics that were used in each experiment. In order to assess further trials, Experiment 1 (originating from YOLOv5s-Seg) is used as the baseline, where no improvement tactics were used. It can be useful to comprehend the effects of various combinations of these tactics on the model’s performance through four separate tests.

The results of the ablation experiments are presented, showing an increasing trend in segmentation and detection accuracy with the gradual addition of improvement strategies. In the meantime, the model’s parameters increase somewhat in comparison to the initial model. This pattern shows how the MLCA attention mechanism and ASF can improve model performance. The accuracy gains for each category in the ablation trials are listed in Table 8.

From Table 9, it can be observed that compared to YOLOv5s-Seg itself, ${mAP}_{bbox}^{50}$ is higher by 1.07% and ${mAP}_{seg}^{50}$ is higher by 1.11%. However, the increase in the number of parameters of the model is minimal, which means that memory usage during model runtime remains almost unchanged. Although the computational complexity (about 9.7%) and inference speed of the model have slightly increased (about 1.7 ms) compared to the original model, considering the accuracy improvement it brings and the fact that it can still meet the needs of rapid evaluation tasks, the improvement method in this paper is satisfying. At the same time, for tasks that focus on building positioning and damage classification, an improved model with only the MLCA attention mechanism (Experiment 2) can be chosen. This model exhibits the highest ${mAP}_{bbox}^{50}$ accuracy and the parameters, computational complexity, and inference time of the model are basically the same as the original model, while the accuracy is significantly improved.

Due to the specificity of post-earthquake rescue operations, people are more concerned about the performance of the model in each category of building damage severity. For example, in real-time rescue operations, more manpower and resources can be allocated to areas with higher degrees of damage, while resources can be saved in undamaged areas, greatly improving rescue efficiency. The model performs best in the Intact and Collapse categories, with ${mAP}_{bbox}^{50}$ and ${mAP}_{seg}^{50}$ in the intact category increasing by 2.73% and 2.73%, respectively, and increasing by 2.58% and 2.14% in the collapse category, respectively. These modifications improve the performance of the model for applications requiring high precision and computational efficiency, while also enhancing the accuracy of categories of interest in rescue missions, making it an ideal solution. Figure 12 illustrates the comparison between the improved YOLOv5s-Seg model and the original model’s prediction results.

Although this study has advanced and the model has demonstrated some identification capabilities in the experiments, there are still certain obstacles and restrictions that require more investigation. For example, due to the perspective from unmanned aerial vehicle (UAV) images, the roofs of buildings are often in the same plane as the roads, making the extraction of dense buildings challenging. Furthermore, due to variations in regional architectural structures, the features of buildings differ across different regions, thus requiring improvement in the model’s generalization ability. Due to a lack of UAV images from other earthquakes, the model’s generalizability to different types of earthquakes and varied building structures is not thoroughly tested. It is a limitation of this research.

Future research will mostly concentrate on the following directions due to the aforementioned limitations: adding UAV images from different perspectives to the dataset to obtain more detailed information about buildings, such as wall damage, in order to improve the accuracy of building damage classification; and enlarging the training dataset to include more post-disaster UAV images of buildings in different types of earthquakes and building structures in different regions, which will improve the model’s identification abilities and increase the model’s applicability. In order to improve overall rescue efficiency, this research will investigate adaptive learning processes in the future. This will make the model more resilient and adaptable to highly dynamic, complex, and challenging environments, including building structures.

## 5. Conclusions

This paper addresses the classification problem of identifying damaged building categories in post-earthquake UAV images and proposes an improved YOLOv5s-Seg instance segmentation algorithm. This research includes activities such as UAV image data collection, data annotation, comparative experiments, and ablation studies.

Several enhancements are introduced to the YOLOv5s-Seg model, including the MLCA attention mechanism and ASF small object detection enhancement in the Neck part. Ablation experiments demonstrate that these enhancement strategies gradually improve the mAP value compared to the original model, while also improving performance on the Intact and Collapse categories, providing valuable decision support for resource allocation during rescue operations.

This research uses comparative trials with other widely used instance segmentation algorithms to demonstrate the superiority of the improved YOLOv5-Seg model in post-earthquake building damage assessment. The improved model outperforms the current methods and can meet the demands of real-time rescue missions with fewer model parameters, higher accuracy, and faster inference speed. Overall, this study thoroughly examines how to apply and improve the YOLOv5s-Seg instance segmentation algorithm for the evaluation of building damage from UAV images following earthquakes, as well as how this improved algorithm can offer algorithmic approaches to post-earthquake rescue operations.

## Figures and Tables

**Figure 1 sensors-24-04371-f001:**
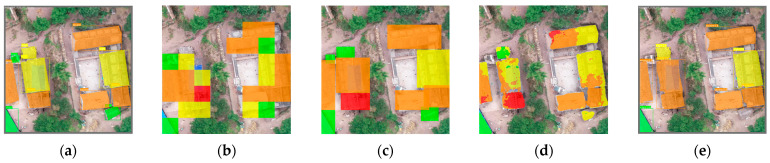
Refer to the legend in Figure 3c for the following categories: green represents the intact category, yellow represents the slight category, orange represents the severe category, and red represents the collapse category. Comparing results from different research methods. (**a**) Label image, real damaged area, and damage category of the building. (**b**) Dividing the image into multiple sub-images and performing classification on each sub-image. (**c**) Object detection, detecting the location of buildings, marking them with rectangles, and classifying the degree of damage. (**d**) Semantic segmentation, classifying each pixel in the image to obtain an overall evaluation. (**e**) Instance segmentation, locating each building and classifying its degree of damage.

**Figure 2 sensors-24-04371-f002:**
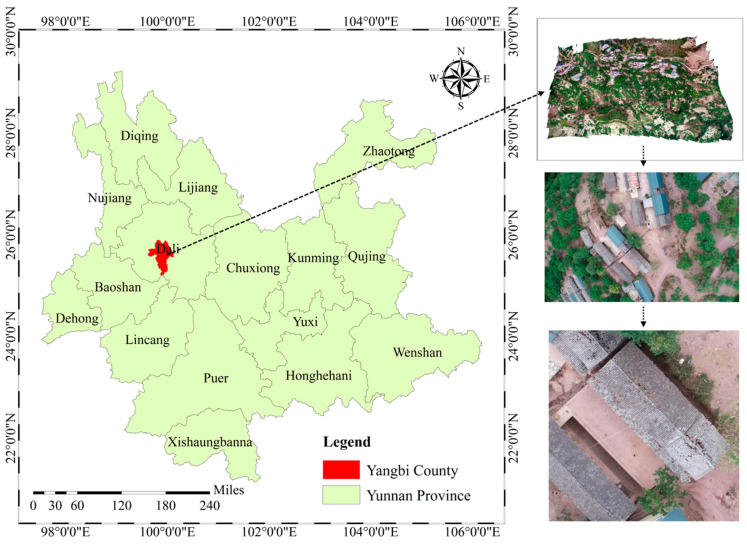
Research area, vector data source: https://geo.datav.aliyun.com/areas_v3/bound/530000_full.json (accessed on 6 June 2024) and https://geo.datav.aliyun.com/areas_v3/bound/532922.json (accessed on 6 June 2024).

**Figure 3 sensors-24-04371-f003:**
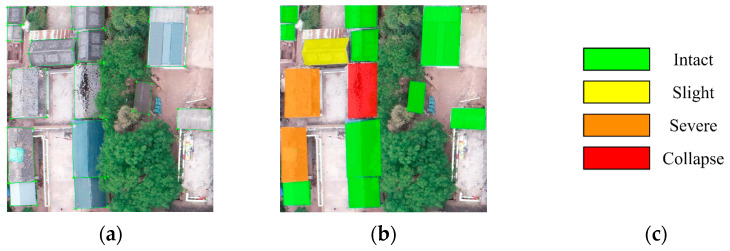
(**a**) Label annotation, depicting the edges of the building. (**b**) Visualization results, marking the damage status of the building. (**c**) Legend of the visualization results, representing different degrees of damage to the building.

**Figure 4 sensors-24-04371-f004:**
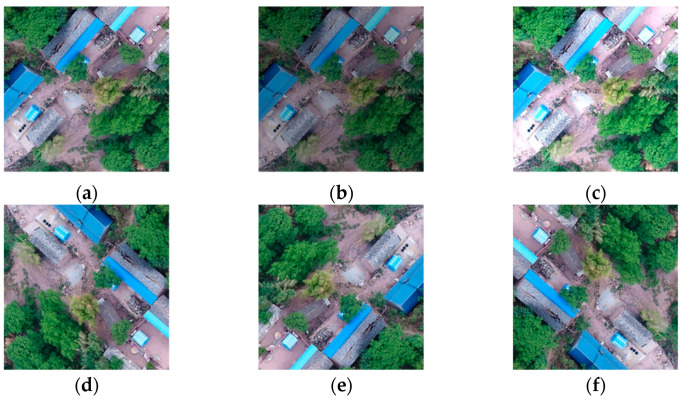
Data augmentation processes. (**a**) Original image, (**b**) brightness −20%, (**c**) brightness +20%, (**d**) rotate right $90^{{}^{\circ}}$, (**e**) rotate right $180^{{}^{\circ}}$, (**f**) rotate right $270^{{}^{\circ}}$.

**Figure 5 sensors-24-04371-f005:**
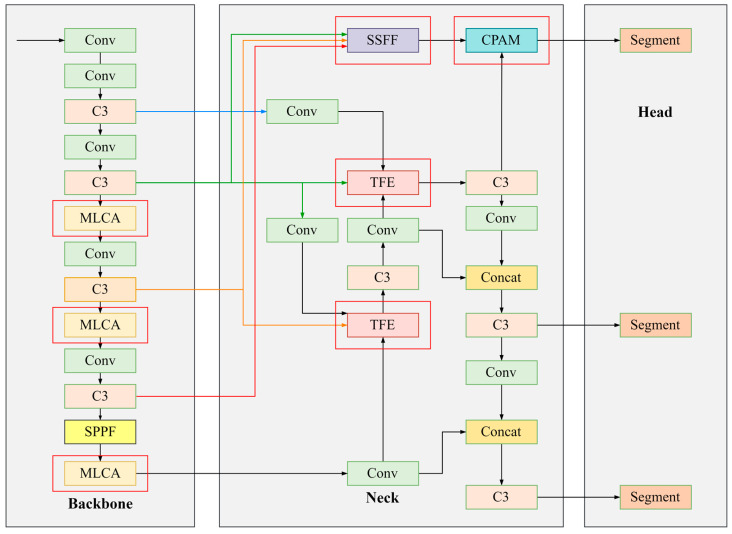
The overall structure of the improved YOLOv5-Seg model. The red rectangular box is the improved part that this paper showcases.

**Figure 6 sensors-24-04371-f006:**
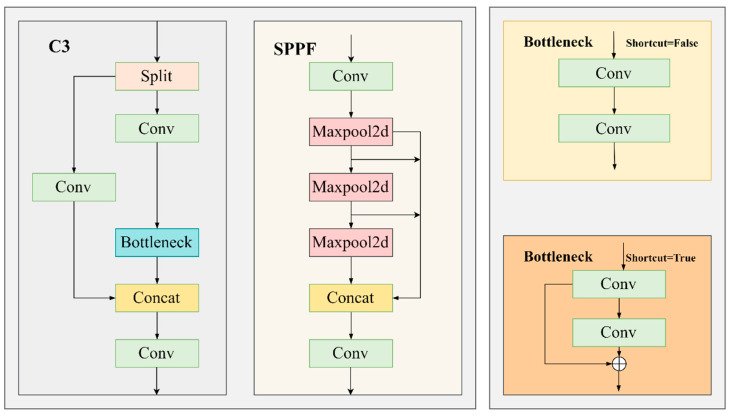
Detailed composition of modules in the improved YOLOv5-Seg model.

**Figure 7 sensors-24-04371-f007:**
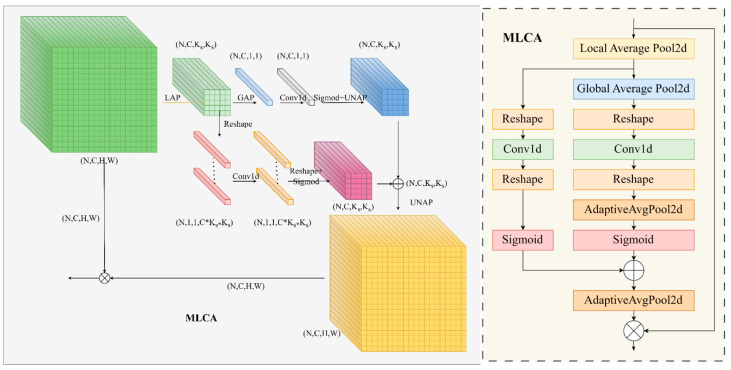
Detailed information about MLCA (Mixed Local Channel Attention).

**Figure 8 sensors-24-04371-f008:**
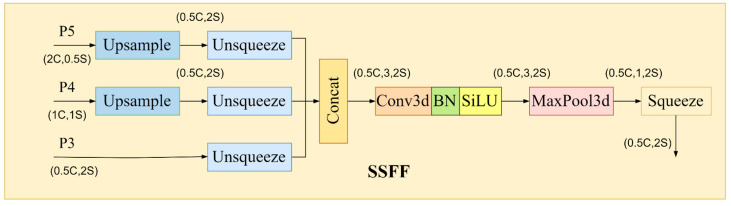
Scale sequence feature fusion (SSFF) module details.

**Figure 9 sensors-24-04371-f009:**
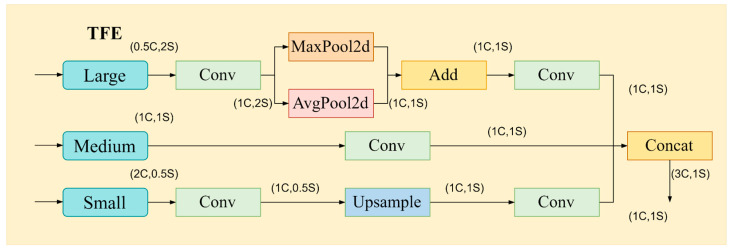
Triple feature encoding (TFE) module details, used for aggregating multi-scale features.

**Figure 10 sensors-24-04371-f010:**
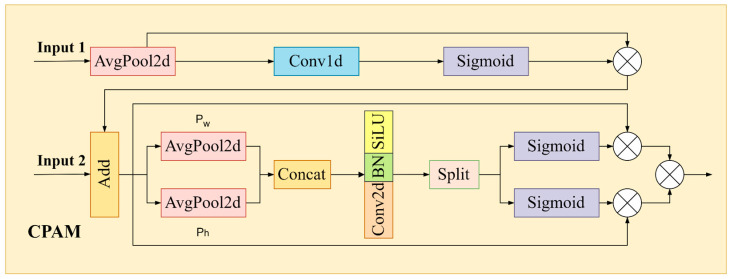
Channel and Position Attention Mechanism (CPAM) module details.

**Figure 11 sensors-24-04371-f011:**
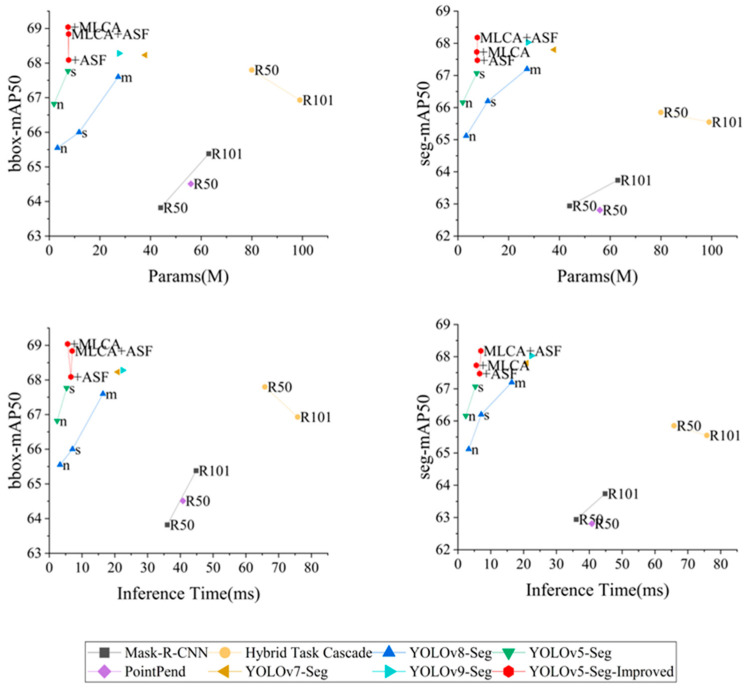
All experimental results were compared in terms of ${mAP}_{bbox}^{50}$ and ${mAP}_{seg}^{50}$ accuracy, parameters, and inference time between different models.

**Figure 12 sensors-24-04371-f012:**
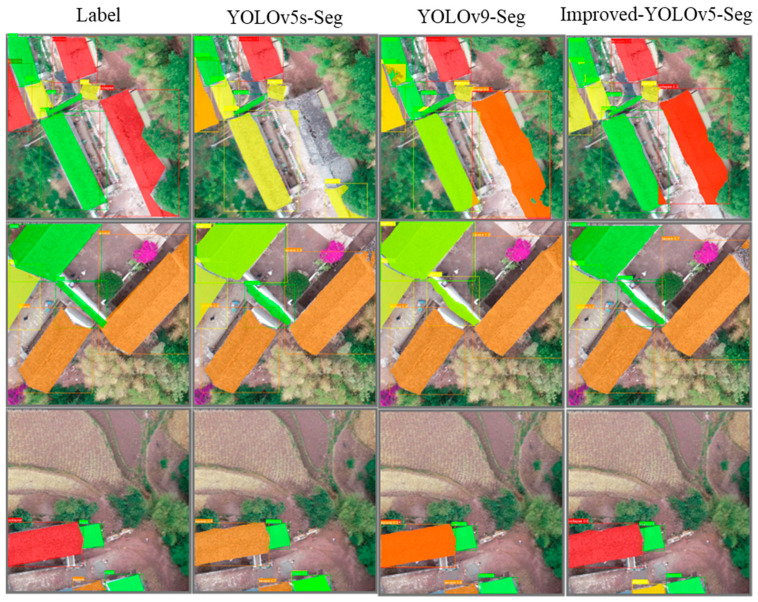
Refer to the legend in Figure 3c for the following categories: green represents the intact category, yellow represents the slight category, orange represents the severe category, and red represents the collapse category. Visualization results comparing the differences in the actual and prediction results of several excellent models. The name of each model is at the top.

**Table 1 sensors-24-04371-t001:** Labels and descriptions.

Class	Example	Description
Intact	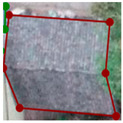	The building outlines are complete, with clear edges and regular geometric shapes. There are no significant textures or abrupt changes in color tone around the buildings. The roofs show no obvious signs of damage, and the overall color transitions naturally.
Slight	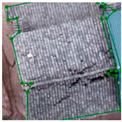	The building outlines are intact, and the roofs and walls are in relatively good condition. However, there are areas where roof tiles have fallen off, resulting in a leaky roof and uneven color tones.
Severe	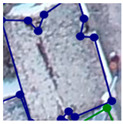	The building outlines are relatively complete, but on one side of the house, there are partial ruins formed due to the collapse of some walls. This debris accumulation exhibits noticeable changes in brightness and color tones in the image.
Collapse	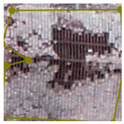	The building outlines are incomplete, and there is asymmetry in the roof texture and color tones. At the collapsed corner of one side of the wall, there is a noticeable contrast with the roof texture and color tones.

**Table 2 sensors-24-04371-t002:** The number of instances for buildings with different levels of damage.

	Images	Intact	Slight	Severe	Collapse
Train Set	4368	8232	6012	3900	2010
Validation Set	1026	1860	1224	924	564

**Table 3 sensors-24-04371-t003:** Experimental Environment.

Server	Details
CPU	12th Gen Intel(R) Core (TM) i9-12900K@3.20 GHz
GPU	NVIDIA GeForce RTX 3090 (24576MiB)
Operation System	Windows11
Memory	64 GB
Pytorch	Version 2.0.1+cu117
Python	Version 3.8.18

**Table 4 sensors-24-04371-t004:** Training settings.

Image Input Size	Training Epoch	Optimizer	Batch Size
1024 × 1024	150	SGD	10

**Table 5 sensors-24-04371-t005:** Information on different YOLOv5-Seg models.

Model	Depth	Width	Parameters (M)
YOLOv5n-Seg	0.33	0.25	1.89
YOLOv5s-Seg	0.33	0.50	7.42
YOLOv5m-Seg	0.67	0.75	21.68
YOLOv5l-Seg	1.00	1.00	47.52
YOLOv5x-Seg	1.33	1.25	88.31

**Table 6 sensors-24-04371-t006:** Detailed results data of all models.

Models	Backbone or Scale	${mAP}_{bbox}^{50}$(%)	${mAP}_{seg}^{50}$(%)	Params (M)	T (ms/img)	FPS (img/s)
Mask R-CNN	ResNet50	63.82	62.94	43.99	36.0	27.8
	ResNet101	65.38	63.74	62.98	44.8	22.3
Hybrid Task Cascade	ResNet50	67.80	65.85	79.97	65.8	15.2
	ResNet101	66.93	65.55	98.96	75.8	13.1
Point Rend [43]	ResNet50	64.51	62.82	55.92	40.8	24.5
YOLOv5-Seg	n	66.82	66.16	1.89	2.4	416.7
	s	67.77	67.07	7.42	5.3	188.7
YOLOv7-Seg	-	68.23	67.80	37.86	20.9	47.9
YOLOv8-Seg	n	65.55	65.12	3.26	3.3	303.0
	s	66.00	66.20	11.78	7.1	140.9
	m	67.60	67.20	27.23	16.4	61.0
YOLOv9-Seg	-	68.28	68.03	27.58	22.4	44.6
Improved YOLOv5-Seg	s	68.84	68.18	7.66	7.0	142.9

**Table 7 sensors-24-04371-t007:** Ablation experiment, “—” means the module has not been added, “✓” means the module has been added.

Different YOLOv5-Seg Algorithm Models	MLCA	ASF
Experiment 1(Baseline)	—	—
Experiment 2	✓	—
Experiment 3	—	✓
Experiment 4	✓	✓

**Table 8 sensors-24-04371-t008:** Ablation experiment results by class.

Improvement	Class	P (%)	R (%)	${mAP}_{bbox}^{50}$(%)	P (%)	R (%)	${mAP}_{seg}^{50}$ (%)
Baseline	Intact	74.67	72.69	78.37	74.74	72.55	77.90
	Slight	56.65	55.64	58.06	56.61	54.98	57.62
	Severe	58.90	62.55	61.16	59.16	62.55	60.02
	Collapse	64.20	70.21	73.49	63.91	69.68	72.73
	Avg	63.61	65.27	67.77	63.61	64.94	67.07
+MLCA	Intact	72.29	74.41	80.34	72.09	74.29	79.95
	Slight	53.81	60.82	56.99	53.39	60.65	56.74
	Severe	59.87	66.04	**63.93**	58.85	65.15	61.72
	Collapse	64.70	73.12	74.88	62.77	71.10	72.51
	Avg	62.67	68.60	69.04	61.78	67.80	67.73
+ASF	Intact	76.12	73.87	80.00	75.80	73.60	78.92
	Slight	58.91	57.35	**58.87**	58.65	57.27	**58.56**
	Severe	61.10	65.58	63.25	60.51	65.15	**61.80**
	Collapse	65.79	71.45	70.26	65.91	71.63	70.60
	Avg	65.48	67.07	68.10	65.22	66.91	67.47
MLCA + ASF	Intact	75.29	75.02	**81.10**	76.81	72.46	**80.63**
	Slight	53.96	57.27	56.60	56.21	53.59	56.11
	Severe	57.85	62.77	61.60	59.96	58.23	61.11
	Collapse	64.13	69.50	**76.07**	64.69	67.55	**74.87**
	Avg	62.81	66.14	68.84	64.42	62.96	68.18

**Table 9 sensors-24-04371-t009:** Ablation experiment result.

Models	${mAP}_{bbox}^{50}$	${mAP}_{seg}^{50}$	Params (M)	FLOPs (G)	T (ms/img)
Experiment 1	67.77	67.07	7.42	25.9	5.3
Experiment 2	**69.04**	67.73	7.42	26.3	5.6
Experiment 3	68.10	67.47	7.66	28.1	6.6
Experiment 4	68.84	**68.18**	7.66	28.4	7

## Data Availability

Data are contained within the article.

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
