# Peer review of "An Improved Instance Segmentation Method for Fast Assessment of Damaged Buildings Based on Post-Earthquake UAV Images"

_sensors, 2024, doi:10.3390/s24134371_

Round 1

Reviewer 1 Report

Comments and Suggestions for Authors

The paper presents an enhanced instance segmentation model designed to quickly and accurately assess building damage from post-earthquake UAV imagery. The proposed model, which improves upon the YOLOv5-Seg architecture, integrates an attention mechanism in the backbone and adopts the ASF-YOLO Neck part for better feature extraction of small targets. The results demonstrate that the improved model outperforms existing methods in both accuracy and efficiency, particularly in classifying different levels of building damage.

The paper provides a thorough comparison with several state-of-the-art models, including Mask R-CNN and different versions of YOLO, demonstrating clear advantages in both performance metrics and speed. Also, the use of UAV imagery from a real earthquake scenario (Yangbi earthquake) adds practical relevance to the study, highlighting the model's potential for real-world applications with some limitations I mentioned below.

However,
1- the study primarily uses data from a single earthquake event. While the results are promising, the model's generalizability to different types of earthquakes and varied building structures is not thoroughly tested. So, at least give some reasoning to clarify this concern, if adding some other data is not possible.
2-
While the improvements in performance are clear, the added complexity from integrating MLCA and ASF mechanisms could be a barrier to wider adoption, especially in resource-constrained environments. Provide a more detailed analysis of the computational trade-offs introduced by the new mechanisms to help practitioners understand the practical implications of adopting the enhanced model.So, the discussion could benefit from a more detailed analysis of the trade-offs involved in terms of computational cost and accuracy.

3-The paper mentions that undamaged samples dominate the learning process, which could skew the model's performance. More emphasis on handling class imbalance would strengthen the study. Therefore, authors can investigate techniques to handle class imbalance more effectively, such as data augmentation or specialized loss functions etc.

Comments on the Quality of English Language

As far as I read, there is nothing to worry about language. 

Reviewer 2 Report

Comments and Suggestions for Authors

Overall, the manuscript entitled "An Improved Instance Segmentation Method for Fast Assessment of Damaged Buildings Based on Post-Earthquake UAV 3 Images" is well-written, and the topic is highly relevant and likely to attract a large audience interested in post-disaster response and building damage assessment.

I have two minor suggestions for improvement:

11. On page 2, the term "data-driven" is misspelled as "da-ta-driven."

22. Some figures, specifically Figures 1 and 12, lack labels. I suggest adding them for clarity and to help readers understand the visual data better.

Reviewer 3 Report

Comments and Suggestions for Authors

This study did interesting and valuable research, which is worthy of publication. The main concerns are as follows.

There is a lot of abbreviations lacing the explanation. Please check throughout the paper and add the corresponding explanation. For example, the full spelling of the abbreviation “UAV” in line 13 is needed.

It is generally not suggested to use pronouns like “ We” and Ietc. in the manuscript.

Line 19, the meaning of mAP... is needed.

As stated in the abstract, the advantage of your method is very limited. It could not highlight the merit of your method.

In Figure 2, the city name should be “Zhaotong” rather than “Shaotong”. Where did you get this map? The original reference is needed.

What is IoU? How to calculate the value of IoU?

Line 241, what are the positive predictions? Please define it clearly and give more details.

This study compares different segmentation methods. However, the training and test procedures of each method were missed in this study, which is suggested to be added.

In addition to the roof damage, damage to walls or columns was widely reported in previous earthquake events. Why did you select to identify the roof damage in the current study?

Round 2

Reviewer 3 Report

Comments and Suggestions for Authors

The author has addressed all of my concerns.